

# The risk of smoking on multiple sclerosis: a meta-analysis based on 20,626 cases from case-control and cohort studies

Peng Zhang, Rui Wang, Zhijun Li, Yuhan Wang, Chunshi Gao, Xin Lv, Yuanyuan Song and Bo Li

Department of Epidemiology and Biostatistics, Jilin University School of Public Health, Changchun, China

## ABSTRACT

**Background.** Multiple sclerosis (MS) has become a disease that represents a tremendous burden on patients, families, and societies. The exact etiology of MS is still unclear, but it is believed that a combination of genetic and environmental factors contribute to this disease. Although some meta-analyses on the association between smoking and MS have been previously published, a number of new studies with larger population data have published since then. Consequently, these additional critical articles need to be taken into consideration.

**Method.** We reviewed articles by searching in PubMed and EMBASE. Both conservative and non-conservative models were used to investigate the association between smoking and the susceptibility to MS. We also explored the effect of smoking on the susceptibility to MS in strata of different genders and smoking habits. The association between passive smoking and MS was also explored.

**Results.** The results of this study suggest that smoking is a risk factor for MS (conservative model: odds ratio (OR) 1.55, 95% CI [1.48–1.62], $p < 0.001$; non-conservative model: 1.57, 95% CI [1.50–1.64], $p < 0.001$). Smoking appears to increase the risk of MS more in men than in women and in current smokers more than in past smokers. People who exposed to passive smoking have higher risk of MS than those unexposed.

**Conclusion.** This study demonstrated that exposure to smoking is an important risk factor for MS. People will benefit from smoking cessation, and policymakers should pay attention to the association between smoking and MS.

## INTRODUCTION

Multiple sclerosis (MS) is an inflammatory disease that occurs when the spinal cord and the insulating covers of the nerve cells in the brain are damaged. This damage affects the nervous system's ability to communicate, resulting in a number of physical and mental problems (*Compston & Coles, 2002*; *Compston & Coles, 2008*). Evidence indicates that MS is an autoimmune disease that directly affects the central nervous system (CNS) myelin or oligodendrocytes. A variety of neurological signs and symptoms are determined by

Corresponding author
Bo Li, li_bo@jlu.edu.cn

the distribution of white matter lesions in the nervous system that may occur in sudden attacks or build up over time (*Compston & Coles, 2008*).

In 2013, there were about 1.5 million people who suffered from MS around the world, with rates varying widely in different regions and populations (*WHO, 2013*). 19,800 people died from MS in 2013, a statistic that was up from 12,400 people in 1990 (*Collaborators, 2015*). The disease usually occurs between the ages of 20 and 50, occupying the leading position of disability among young adults. The risk of MS for females is two times as high as males (*Milo & Kahana, 2010*).

The cause of MS is still not clear, but through rigorous epidemiological investigation, genetic variations, the Epstein–Barr virus infection, vitamin D nutrition and cigarette smoking have been identified as likely causal factors for MS (*Handel et al., 2010*; *Ramagopalan et al., 2009*; *Simon et al., 2012*).

A previous meta-analysis published in 2014 reported a pooled odds ratio (OR) of 1.51 (95% CI [1.38–1.65]) for the association between smoking and MS susceptibility (*O'Gorman & Broadley, 2014*). However, the evidence was suggestive rather than sufficient about the role of smoking in the etiology of MS because the sample sizes were relatively small. Many recent studies have explored the association between smoking and MS either directly or indirectly. Therefore, we conducted this meta-analysis to investigate the association in a larger sample. Moreover, we aimed to detect the effect of smoking on the incidence of MS in strata of different genders and smoking habits.

## MATERIALS AND METHODS

### Search strategy

We identified published studies that explored the association between smoking and the risk of MS by searching the PubMed and EMBASE databases from January 1st, 1980 to March 31st, 2015. The following search terms were used: "multiple sclerosis," "case-control," "cohort study," "birth cohort," "survival analysis," "cigarette smoking," "tobacco smoking" and "cigars." In addition, the reference list of retrieved papers was also reviewed to identify additional relevant studies.

### Selection criteria

The eligible studies needed to meet the following criteria: (1) the study must be an original study, (2) the study must investigate the association between smoking and the incidence of multiple sclerosis, (3) the study must include at least 50 cases, and (4) the study must report the odds ratio (OR), relative risk (RR) with its corresponding 95% confidence interval (95% CI), or the number of events to calculate them.

### Study selection and data extraction

The articles retrieved from the database were independently evaluated by two reviewers (Peng Zhang and Rui Wang) based on the aforementioned selection criteria. Studies designed as systematic review and duplicate studies of the same population were excluded. Articles that contained multiple study populations were divided into separate studies. Disagreements were resolved by discussion. Articles in which disagreements could not

be resolved were all included. The following information were extracted from the eligible studies: first author, year of publication, country of origin, OR or RR with its 95% CI, study design, the method of information collection, method of MS diagnosis, and the relationship between disease onset and the duration of smoking.

## Statistical analysis

The rare disease assumption was used to combine the odds ratio (OR) and relative risk (RR) (*Clayton & Hills, 1993*). If the RR or OR and its 95% CI were not reported but sufficient information was available, we used previously described methods to calculate it (*Bland & Altman, 2000*). Stata12.0 was used to compute the pooled ORs and their 95% CI, to generate forest plots and to assess the heterogeneity of the included studies. As described in the former meta-analysis (*Handel et al., 2011*), we also performed this meta-analysis using conservative (including only studies where smoking behavior was described prior to disease onset) and non-conservative (all studies regardless of whether smoking behavior occurred before onset or concurrently) models. To test the stability of the results, we investigated the influence of a single study on the overall effect value by removing one study each time. ORs were calculated among the subgroups of studies and compared across them. Possible publication bias was assessed using Begg's funnel plot and Egger's test (*Begg & Mazumdar, 1994*; *Egger et al., 1997*).

# RESULTS

## Search result and study characteristics

After selecting studies according to the inclusion criteria, 47 articles considered for further review. Six of these 47 articles could not provide outcome information (*Brosseau et al., 1993*; *Guimond et al., 2014*; *Lauer, 2006*; *Nortvedt, Riise & Maeland, 2005*; *Senecal-Quevillon, Duquette & Richer, 1986*; *Turner et al., 2007*). We could not obtain the full article for 5 of the 47 articles (*Dobosz, Tyrpien & Pierzchala, 2012*; *Frutos Alegria et al., 2002a*; *Frutos Alegria et al., 2002b*; *Ragonese et al., 2007*; *Rodriguez Regal et al., 2009*). Ten of these 47 articles contained duplicate study populations (*Baarnhielm et al., 2012*; *Hedstrom et al., 2011a*; *Hedstrom et al., 2009*; *Hedstrom et al., 2013a*; *Hedstrom et al., 2014b*; *Hedstrom et al., 2011b*; *Munger, Chitnis & Ascherio, 2009*; *Munger et al., 2003*; *Sundqvist et al., 2012*; *Sundstrom, Nystrom & Hallmans, 2008*). Ultimately, 26 eligible articles containing 29 study populations were identified (*Al-Afasy et al., 2013*; *Alonso et al., 2011*; *Asadollahi et al., 2013*; *Briggs et al., 2014*; *Carlens et al., 2010*; *Ghadirian et al., 2001*; *Gustavsen et al., 2014*; *Hedstrom et al., 2013b*; *Hernan et al., 2005*; *Hernan, Olek & Ascherio, 2001*; *Jafari et al., 2009*; *Kotzamani et al., 2012*; *Maghzi et al., 2011*; *Mansouri et al., 2014*; *O'Gorman et al., 2014*; *Pekmezovic et al., 2006*; *Ragnedda et al., 2015*; *Ramagopalan et al., 2013*; *Riise, Nortvedt & Ascherio, 2003*; *Russo et al., 2008*; *Silva et al., 2009*; *Simon et al., 2015*; *Simon et al., 2010*; *Thorogood & Hannaford, 1998*; *Villard-Mackintosh & Vessey, 1993*; *Zorzon et al., 2003*). A flow chart for the study selection process was shown in Fig. 1. There were 19,834 cases of MS and 21,350 controls in case-control studies; 792 cases of MS occurred in 601,492 individuals in cohort studies. Among these studies, four were conducted in Iran, four in America, three in England, three in Norway, two in Canada,

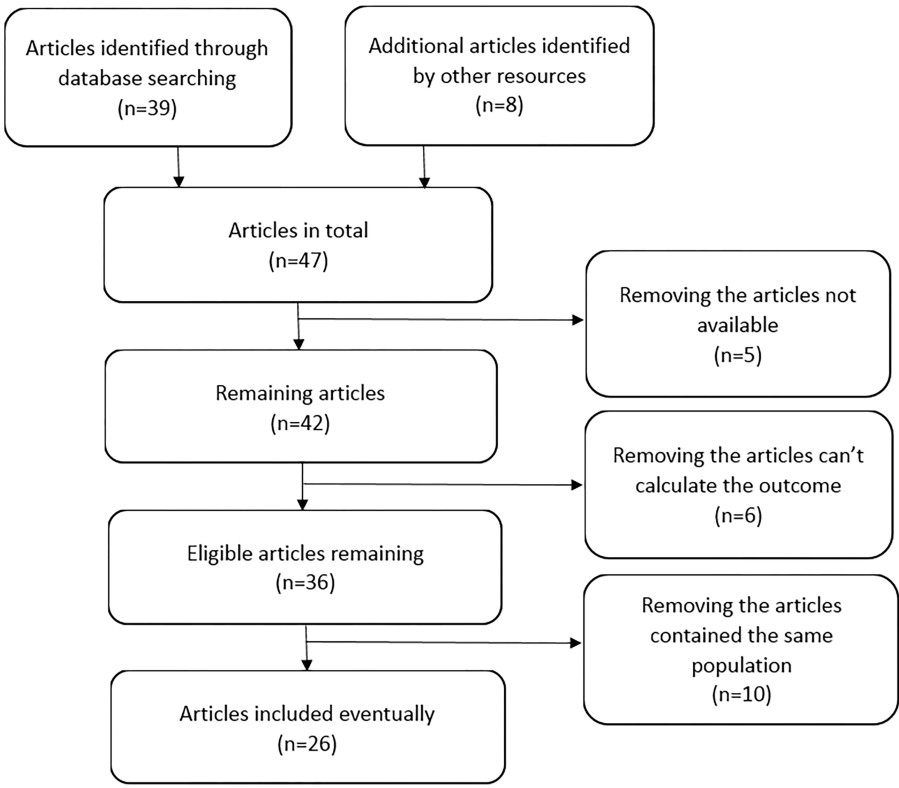

**Figure 1** Summary of the studies selection process.

three in Sweden, one in Brazil, one in Greece, two in Australia, one in the Netherlands, one in Kuwait, three in Italy, one in Serbia. The main characteristics of the included studies are summarized in Table 1.

## Smoking and MS susceptibility

The conservative model contained 24 studies that investigated the association between smoking and MS. Moderate heterogeneity was detected ($I^2 = 37.2\%$, $p = 0.035$). As described in Fig. 2, the pooled OR was 1.55 (95% CI [1.48–1.62], $p < 0.001$), indicating that ever-smoking increases the risk of MS by 55% compared with never-smoking individuals. When including all 29 studies in the non-conservative model, we obtained similar results (OR = 1.57, 95% CI [1.50–1.64], $p < 0.001$, heterogeneity: $I^2 = 47.3\%$, $p = 0.003$; Fig. 3). There were no significant differences among the subgroups based on study designs, diagnostic criteria, or the data collection methods; however, not adjusting for confounders may overestimate the risk between smoking and MS susceptibility (Table 2).

## Different effects of genders and smoking habits

In total, 10 studies provided enough information to report the association between smoking and MS within genders (*Asadollahi et al., 2013*; *Carlens et al., 2010*; *Hedstrom et al., 2009*; *Hernan, Olek & Ascherio, 2001*; *Kotzamani et al., 2012*; *Maghzi et al., 2011*;

Zhang et al. (2016), *PeerJ*, DOI 10.7717/peerj.1797

Peerj

**Table 1** The main characteristics of the included studies.

| 1st author and year of publication | Cases | Controls or observational individual | OR or RR(95% CI) versus never-smoking | Information collecting | Type | Diagnostic criteria | Smoking and the onset of MS |
|---|---|---|---|---|---|---|---|
| Ragnedda 2015 (Norwegian) | 894 | 1,610 | 2.00(1.68, 2.38) (ever-smoking) | Questionnaire | Case-control | McDonald | Before onset |
| Ragnedda 2015 (Italian) | 617 | 1,161 | 1.55(1.28, 1.88) (ever-smoking) | Questionnaire | Case-control | McDonald | Before onset |
| Simon 2014 | 1,190 | 454 | 1.4(1.1, 1.9) (ever-smoking) | Face interview | Case-control | N/A | Before onset |
| Gustavsen 2014 | 530 | 918 | 2.29(1.82, 2.89) (ever-smoking) | Questionnaire | Case-control | McDonald or Poser | current |
| Mansouri 2014 | 1,217 | 787 | 1.93(1.31, 2.73) (ever-smoking) | Face interview | Case-control | McDonald or Poser | Before onset |
| O'Gorman 2014 | 560 | 480 | 1.9(1.5, 2.5) (ever-smoking) 3.6(2.3, 5.6) (current-smoking) 1.6(1.2, 2.1) (past-smoking) | Questionnaire | Case-control | Physician | Current |
| Briggs 2014 | 1,012 | 576 | 1.27(1.03, 1.58) (ever-smoking) | Telephone questionnaire | Case-control | McDonald | Before onset |
| Asadollahi 2013 | 662 | 394 | 1.78(1.22, 2.59) (ever-smoking) | Face or telephone interview | Case-control | McDonald or Poser | Before onset |
| Hedström 2013 | 6,990 | 8,279 | 1.49(1.40, 1.59) (ever-smoking) 1.56(1.45, 1.67) (current-smoking) 1.35(1.24, 1.47) (past-smoking) | Questionnaire | Case-control | McDonald | Before onset |
| Ramagopalan 2013 | 3,157 | 756 | 1.32(1.10, 1.60) (ever-smoking) | Questionnaire | Case-control | N/A | Current |
| Kotzamani 2012 | 504 | 591 | 1.9(1.50, 2.41) (ever-smoking) | Questionnaire | Case-control | N/A | Before onset |
| Al-Afasy 2010 | 101 | 202 | 1.7(0.9, 2.4) (ever-smoking) | Face interview | Case-control | Neurologist | Before onset |
| Maghzi 2011 | 516 | 1,090 | 2.67(1.70, 4.21) (ever-smoking) | Questionnaire | Case-control | McDonald | Before onset |
| Alonso 2011 | 394 | 394 | 1.72(0.90, 3.30) (ever-smoking) | Telephone interview | Case-control | McDonald | Before onset |
| Simon 2010a | 210 | 420 | 1.4(1.0, 2.0) (ever-smoking) | Questionnaire | Case-control | N/A | Before onset |

Zhang et al. (2016), *PeerJ*, DOI 10.7717/peerj.1797

Peer J

**Table 1** (*continued*)

| 1st author and year of publication | Cases | Controls or observational individual | OR or RR(95% CI) versus never-smoking | Information collecting | Type | Diagnostic criteria | Smoking and the onset of MS |
|---|---|---|---|---|---|---|---|
| Simon 2010b | 136 | 272 | 1.5(1.0, 2.4) (ever-smoking) | Interview | Case-control | Poser | Before onset |
| Simon 2010c | 96 | 173 | 1.4(0.8, 2.4) (ever-smoking) | Questionnaire | Case-control | N/A | Before onset |
| Carlens 2010 | 214 | 277,777 | 2.5(1.7, 3.6) (ever-smoking) 2.8(1.9, 4.2) (current-smoking) 1.6(0.9, 2.8) (past-smoking) | N/A | Cohort | N/A | Before onset |
| Jafari 2009 | 136 | 204 | 1.09(0.68, 1.73) (ever-smoking) 1.03(0.61, 1.73) (current-smoking) 1.19(0.65, 2.20) (past-smoking) | Questionnaire | Case-control | McDonald | Before onset |
| Silva 2009 | 81 | 81 | 2.0(0.9, 4.3) (current-smoking) | Face interview | Case-control | Poser | Current |
| Russo 2008 | 94 | 53 | 1.83(0.86, 3.87) (ever-smoking) | N/A | Case-control | McDonald | N/A |
| Pekmezovic 2006 | 196 | 210 | 1.6(1.08, 2.37) (ever-smoking) | Face interview | Case-control | Poser | Before onset |
| Hernan 2005 | 210 | 1,913 | 1.3(1.0, 1.7) (ever-smoking) 1.4(1.0, 1.9) (current-smoking) 1.0(0.6, 1.8) (past-smoking) | Questionnaire | Case-control | Poser | Before onset |
| Riise 2003 | 87 | 22,312 | 1.81(1.13, 2.92) (ever-smoking) | Questionnaire | Cohort | Self-report | Before onset |
| Zorzon 2003 | 140 | 131 | 1.50(0.90, 2.40) (ever-smoking) | Interview | Case-control | McDonald | Before onset |
| Hernan 2001 | 314 | 238,371 | 1.6(1.2, 2.1) (current-smoking) 1.2(0.9,1.6) (past-smoking) | Questionnaire | Cohort | Physician | Before onset |
| Ghadirian 2001 | 200 | 202 | 1.6(1.0, 2.4) (ever-smoking) | Face interview | Case-control | N/A | Before onset |
| Thorogood 1998 | 114 | 46,000 | 1.2(0.8, 1.8) (1–14/day) | N/A | Cohort | Physician | Before onset |
| Villard 1993 | 63 | 17,032 | 1.5(0.6, 3.3) (ever-smoking) | N/A | Cohort | N/A | Before onset |

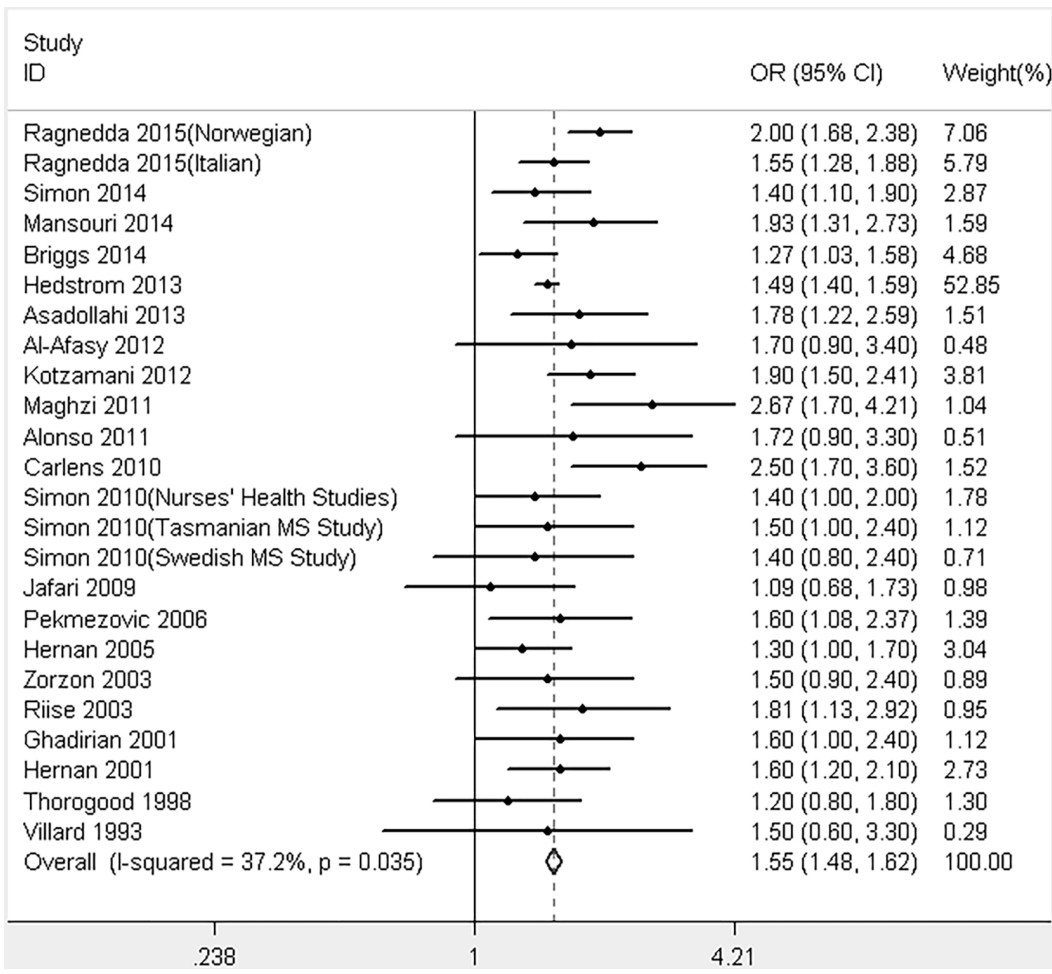

**Figure 2  Forest plot of smoking and multiple sclerosis risk (conservative model).**

**Table 2  Odds ratio and 95% confidence intervals for different subgroups of studies.**

| Subgroups | Number of studies | Odds ratio | 95% CIs | p-value for comparison |
|---|---|---|---|---|
| Case-control | 24 | 1.56 | 1.49–1.63 | 0.362 |
| Cohort | 5 | 1.70 | 1.42–2.03 | |
| McDonald/ Poser criteria | 16 | 1.70 | 1.52–1.90 | 0.124 |
| Physician/self-reported/not reported | 13 | 1.52 | 1.39–1.66 | |
| Adjustment for covariates | 15 | 1.51 | 1.43–1.59 | 0.005 |
| No adjustment | 14 | 1.74 | 1.60–1.89 | |
| Self-administrated questionnaire | 14 | 1.58 | 1.43–1.74 | 0.674 |
| Face or telephone interview/not report | 15 | 1.63 | 1.47–1.82 | |

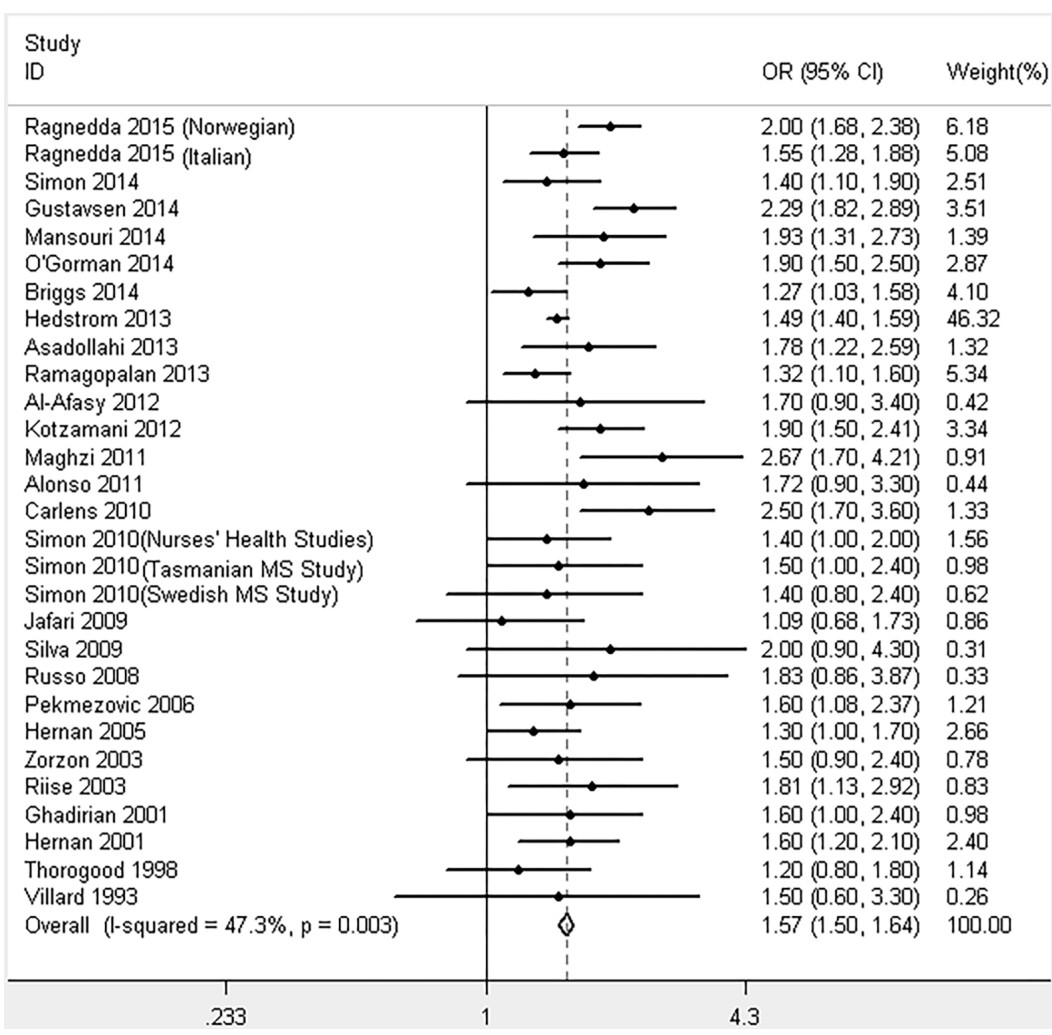

**Figure 3** **Forest plot of smoking and multiple sclerosis risk (non-conservative model).**

*O'Gorman et al., 2014*; *Simon et al., 2010*; *Thorogood & Hannaford, 1998*; *Villard-Mackintosh & Vessey, 1993*). Significant differences were detected between different genders ($\chi^2 = 11.21$, $p = 0.001$, Fig. 4). Smoking in men is more dangerous than women. Similarly, we included 7 studies that provided data about the effects of different smoking habits on susceptibility to MS (*Carlens et al., 2010*; *Hedstrom et al., 2013a*; *Hernan et al., 2005*; *Hernan, Olek & Ascherio, 2001*; *Jafari et al., 2009*; *O'Gorman et al., 2014*; *Zorzon et al., 2003*). Being a current smoker increases the risk of MS by 83% risk compared with nonsmokers; past smoking increases the risk of MS by 58% compared with nonsmokers. Significant differences were detected between the effects of current and past smoking versus non-smokers ($\chi^2 = 12.66$, $p < 0.001$, Fig. 5). In order to explore the impact of passive smoking (active smokers were excluded) on the risk of MS, we identified 3 eligible articles containing four study populations (*Hedstrom et al., 2014a*; *Hedstrom et al., 2013b*;

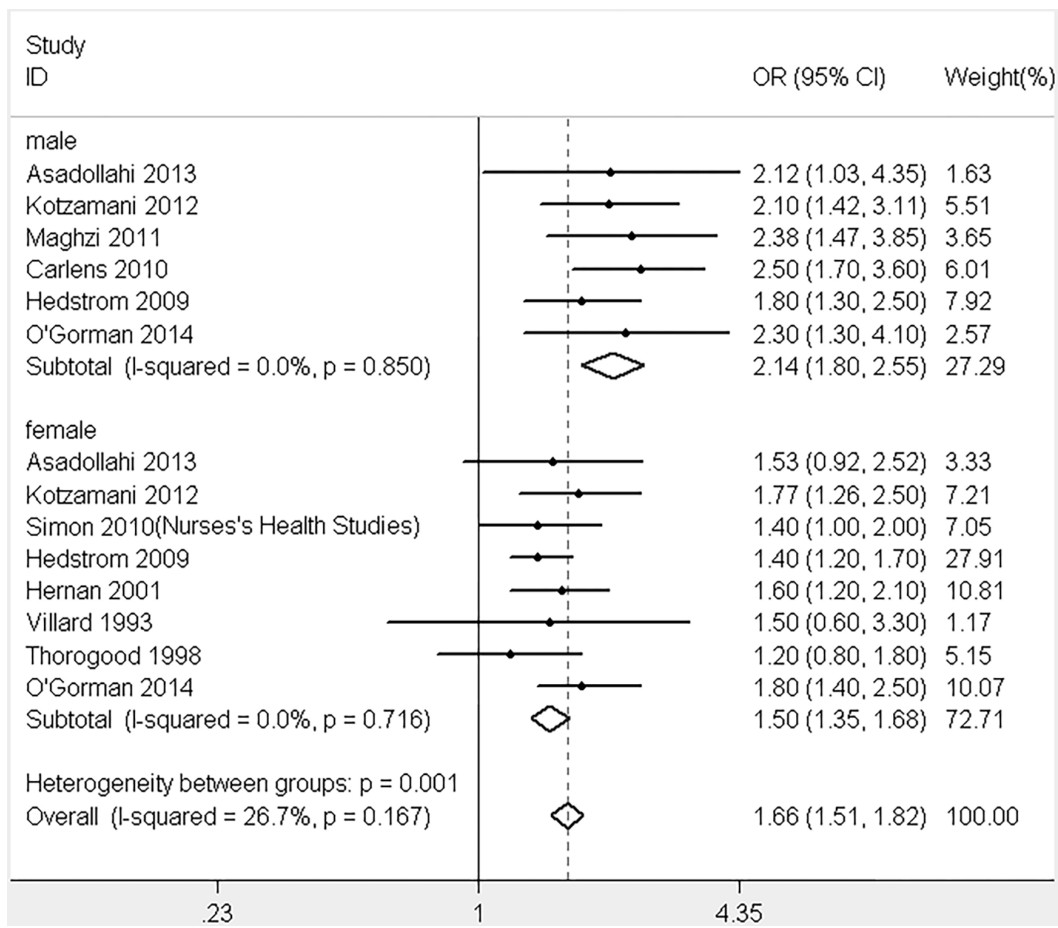

**Figure 4 Forest plot of smoking and risk of multiple sclerosis in different genders.**

*Ramagopalan et al., 2013*). As described in Fig. 6, the pooled OR was 1.24 (95% CI [1.03–1.49], $p = 0.028$), indicating that exposure to passive smoking increases the risk of MS by 24% compared with unexposed individuals.

## Sensitivity analysis and publication bias

Figure 7 implied the funnel plot was symmetrical, suggesting no publication bias. The Begg rank correction test and Egger linear regression showed no asymmetry (Begg, $p = 0.612$; Egger, $p = 0.204$).

Figure 8 showed the result of the sensitivity analysis by removing one study in each turn. This procedure showed that the study by Hedstrom in 2013 significantly impacted the main result. When switched from fixed effects model to random effects model, the OR changed from 1.57 (95% CI [1.50–1.64], $p < 0.001$) to 1.63 (95% CI [1.51–1.76], $p < 0.001$), suggesting that the result was robustness.

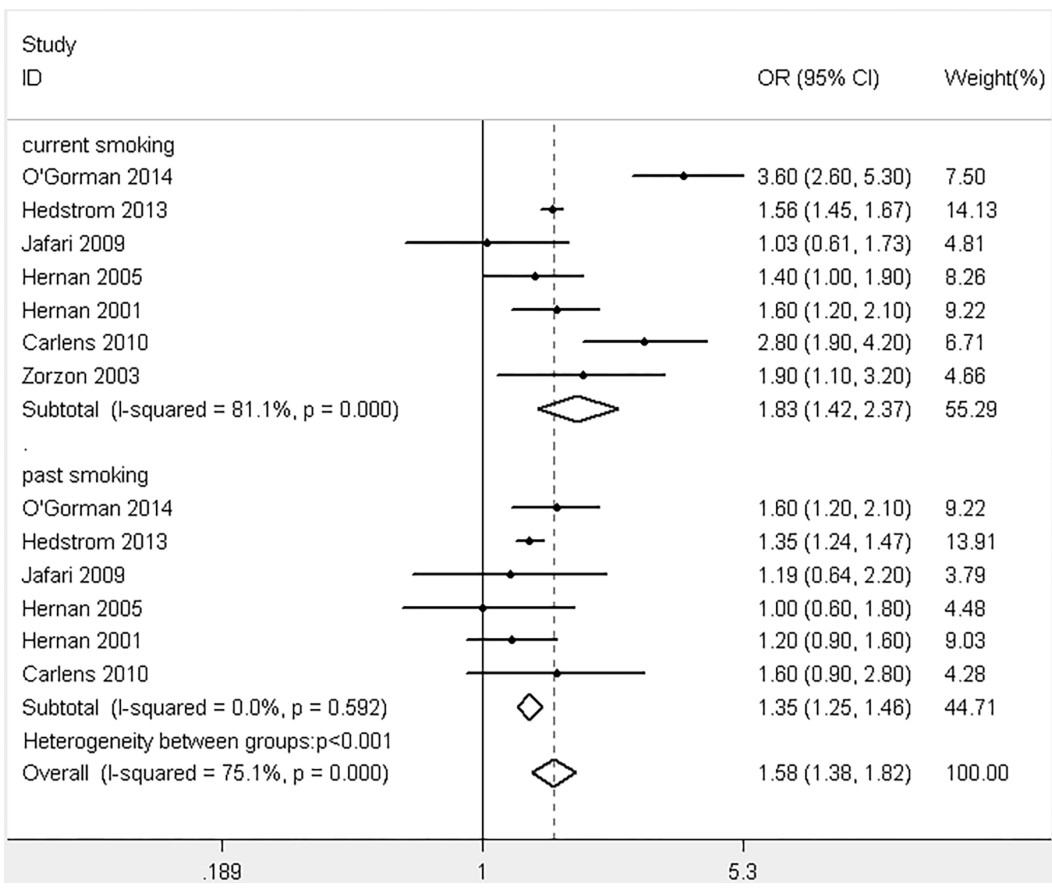

**Figure 5** Forest plot of smoking and risk of multiple sclerosis in different smoking habits.

# DISCUSSION

Our meta-analysis showed there was a strong association between smoking and MS susceptibility. Ever-smoking could increase the risk of MS by a more than 50% risk compared with never-smoking population. The non-conservative model obtained a similar result compared with the conservative model, suggesting a robustness of the results. The subgroup analyses showed that different study designs, diagnostic criteria and types of information resource had little impact on the relationship between smoking and MS susceptibility. However, inadequate adjustment may overestimate the risk between smoking and MS susceptibility. The sensitivity analysis showed the study by Hedstrom 2013 significantly impacted the main result. Therefore, we reviewed this article and found that it included 6,990 cases (no snuff use) and 8,279 controls (no snuff use) that constituted 46.32% of the entire meta-analysis. Male smokers were shown to have a higher risk of developing MS than female, but the exact number of cigarettes consumed by different genders per day due to different lifestyle habits was unavailable, so we were unable to draw a firm conclusion. Significant differences were detected between the effects of current and former smokers compared with non-smokers. Current smoking is more dangerous than past smoking, which informed individuals of the benefits of smoking

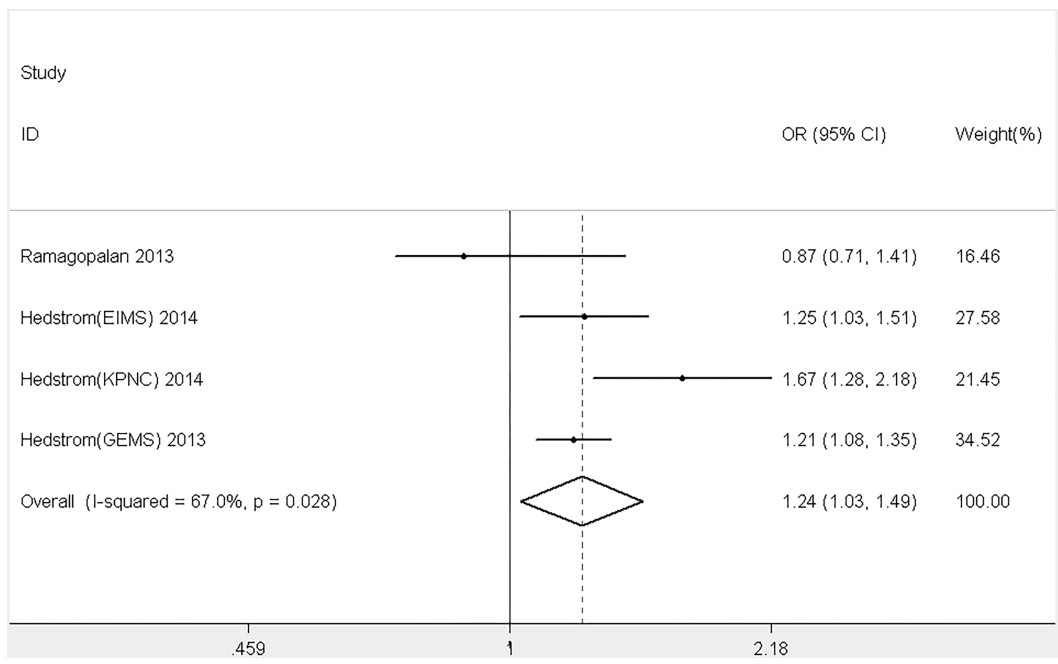

**Figure 6  Forest plot of passive smoking and multiple sclerosis risk.**

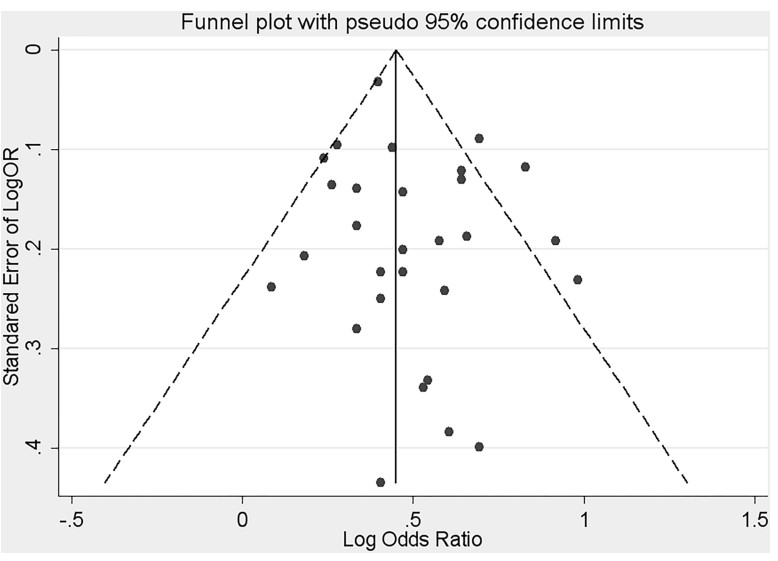

**Figure 7  Funnel plot based on related risk for association between smoking and multiple sclerosis.**

cessation. Passive smoking is a risk factor for MS in non-smoking population. Smoke-free environment in public places and home is vital to people's health.

Comparing with three former meta-analyses (*Hawkes, 2007* (OR = 1.34), Handel 2011 (OR = 1.52), O'Gorman 2014 (OR = 1.51)), our study obtained a greater effect estimates between smoking and MS susceptibility (OR = 1.57) (*Handel et al., 2011*; *Hawkes, 2007*;

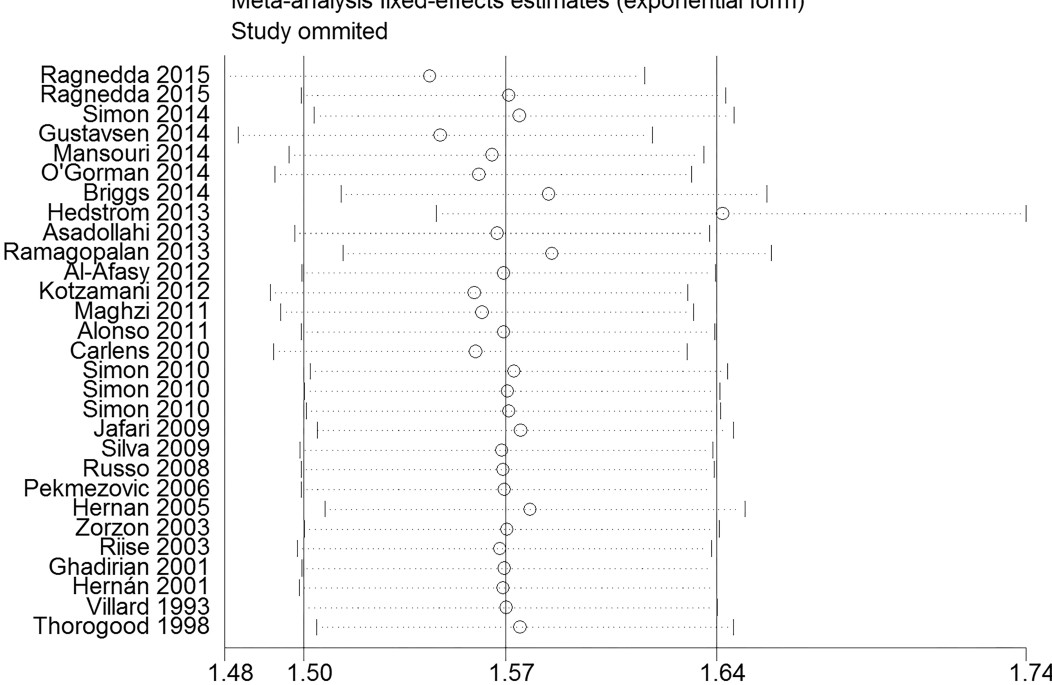

**Figure 8** Forest plot of sensitivity analysis by removing each study in each turn.

*O'Gorman & Broadley, 2014*). Studies published from 2013 to 2015 accounted for 78.62% of the entire meta-analysis and reported higher effect estimates.

The etiology of MS is still unknown, and both genetic and environmental factors may contribute to this disease (*Compston & Coles, 2008*). MS is more common with the increasing latitude, except for some ethnic groups such as the New Zealand Moori (*Pugliatti, Sotgiu & Rosati, 2002*), Canada's Inuit (*Milo & Kahana, 2010*) and inland Sicilians (*Grimaldi et al., 2001*); however, the reasons for these geographical distributions are still controversial (*Milo & Kahana, 2010*). Some people believe that a possible explanation could be that decreased exposure to sunlight results in decreased levels of vitamin D (*Ascherio & Munger, 2007*; *Ascherio, Munger & Simon, 2010*), while others believe that it is a consequence of the distribution of the northern European populations that had a high prevalence of MS (*Milo & Kahana, 2010*). Although MS is not considered to be a hereditary disease, the probability of MS is higher if there is a family history of the disease (*Compston & Coles, 2002*). Differences of specific genes in the human leukocyte antigen (HLA) system that serve as the major histocompatibility complex (MHC) may be associated with MS susceptibility (*Compston & Coles, 2008*).

The causal link between cigarette smoking and MS is still unclear (*Jafari & Hintzen, 2011*). There are more than 4,500 types of possible toxic substances, including nicotine and nitric oxide in cigarette smoke. Some nerve lesions, such as axonal degeneration, have been caused by exposure to nitric oxide (*Scolding & Franklin, 1998*; *Smith, Kapoor & Felts, 1999*). A study in Sweden showed the inhalation of non-nicotinic components of cigarette smoke are more influential than nicotine in the etiology of MS (*Carlens et al.,*

*2010*). This finding suggests the real reason for the elevated risk of MS is the irritation of cigarette smoke in the lungs, triggering the pro-inflammatory effect of smoking via toll-like receptors (*Mortaz et al., 2009*; *Pace et al., 2008*). As a type of lymphocyte, T-cells enter the brain by destroying the blood–brain barrier in the inflammatory process. The T-cell recognized myelin as exogenous material and attacked it, causing the loss of myelin (*Compston & Coles, 2008*). Further damage of the blood–brain barrier will lead to a number of other effects, such as the activation of cytokines and modification of proteins that may break self-tolerance, resulting in autoimmune responses against antigens of the nervous system (*Makrygiannakis et al., 2008*).

Most of the studies included in this meta-analysis focus on the risk of MS between having ever smoked and never smoking; however, the exact dose of cigarette consumption as well as how these data were recorded vary from study to study (pack-years, per day etc.). Therefore, it is difficult to assess the association between the degree of MS susceptibility and the degree of cigarette consumption based on current studies.

## CONCLUSIONS

Our meta-analysis suggests that exposure to smoking is an important risk factor for MS. People would benefit from quitting smoking, and policymakers should pay attention to this association. Further research is needed to assess the dose–response effect between smoking and MS.

### Funding
The authors received no funding for this work.

### Competing Interests
The authors declare there are no competing interests.

### Author Contributions
- Peng Zhang conceived and designed the experiments, performed the experiments, analyzed the data, wrote the paper, prepared figures and/or tables, reviewed drafts of the paper.
- Rui Wang conceived and designed the experiments, analyzed the data, wrote the paper, prepared figures and/or tables.
- Zhijun Li conceived and designed the experiments, analyzed the data.
- Yuhan Wang contributed reagents/materials/analysis tools.
- Chunshi Gao and Xin Lv performed the experiments, contributed reagents/materials/-analysis tools, prepared figures and/or tables, reviewed drafts of the paper.
- Yuanyuan Song performed the experiments, contributed reagents/materials/analysis tools.
- Bo Li conceived and designed the experiments.

## Data Availability

The research in this article did not generate any raw data.

## Supplemental Information

Supplemental information for this article can be found online at http://dx.doi.org/10.7717/peerj.1797#supplemental-information.

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
