# Peer review of "The risk of smoking on multiple sclerosis: a meta-analysis based on 20,626 cases from case-control and cohort studies"

_PeerJ, doi:10.7717/peerj.1797_

## Round 0.1 · original submission · Major Revisions

Dear authors,

Ater analysing the comments provided by the reviewers, both of them have indicated that your manuscript has scientific merit. However, they have also highlighted issues which you should solve before publication.

With respect and warm regards,
Dr Palazón-Bru (academic editor for PeerJ)

·

Basic reporting

when my problem in validityof the findings resolved, I will send the commends in this area.

Experimental design

when my problem in validityof the findings resolved, I will send the commends in this area.

Validity of the findings

I think the most important problem in this study is in the results.the results showed that men and current smokers have higher risks of the disease. almost in all studies showed that incidence of Ms higher in females no males.this result in this study seemed odd and atypical.

Additional comments

I suggest that the dear author have details analysis on gender as a mediator between smoking and MS

Reviewer 2 ·

Basic reporting

This is a fair meta-analysis on the association between smoking and the risk of MS. As the authors state in the introduction, other meta-analyses have been conducted, with similar conclusions. However, there have been a number of new case-control and cohort studies on this subject, justifying a new update.

The statistical calculations appear valid and well performed, and the authors appear to have included all relevant articles in the analysis. The conclusions are justified by the data.

Major points:
1)
The language needs substantially improvement. There are a number of orthographic and spelling errors, even in the title: The risk of smoking on Multiple sclerosis: A meta-analysis based on 20,626 cases from a case-control and cohort study. I would guess that the authors mean to write: (..) from case-control and cohort studies. Some sentences make no sense at all, for example “Conventional wisdom has investigated the association between cigarette smoking and MS”. The orthographic and spelling-errors must be corrected before I can recommend publication.

2)
The authors have not addressed the impact of passive smoking on MS-risk. As a number of epidemiological studies have also addressed this subject, inclusion of these analyses in the meta-analysis would greatly increase the readability and relevance of the data.

Experimental design

No Comments.

Validity of the findings

No Comments.

---

## Round 0.2 · Major Revisions

Dear authors,

Thank you very much for the opportunity to review your paper. I have re-invited the 2 original reviewers and one additional one - all of them have indicated your manuscript has scientific merit to be published in PeerJ, once some remaining issues are solved.

·

Basic reporting

it is seemed generality of manuscript is good.

Experimental design

I have no idea

Validity of the findings

the results of this study especialy about the relationship between gender and Ms is not common and is not coordinating with other majority of studies.

Reviewer 2 ·

Basic reporting

The manuscript has been largely improved and my concerns have been addressed adequately.

I have two final minor concerns:

1) Abstract
In the results section: The Authors Write: "Both smoking habits and genders affect the risk of MS; men and current smokers have higher risks of the disease." This gives the impression that men in general have a higher risk of MS than women. I would have rewritten this sentence as: "smoking appears to increase the risk of MS more in men than in women and in current smokers more than in past smokers."

2) Discussion
I would recommend that authors remove the following section from the discussion: "With the development of modern genetic methods, such as genome-wide association studies, more than 12 other genes outside the HLA have been found to be linked with MS susceptibility(Baranzini 2011). Many infectious agents were also considered to be a trigger of MS, such as Epstein-Barr virus but not have been confirmed(Lucas & Taylor 2012)", since these sentences make little contribution to to overall text and are beyond the scopes of the article.

Experimental design

No Comments

Validity of the findings

No Comments

Additional comments

No Comments

Reviewer 3 ·

Basic reporting

just some comments about (English) writing, which I will point in the general comments with suggestions.

Experimental design

Influence of environmental factors on MS susceptibility is indeed very important. Although this meta-analysis does not give new information or results, still it is interesting because it confirms finding of earlier studies and meta-analysis in a bigger sample size.

Validity of the findings

No comments

Additional comments

- add space before open parenthesis. Check the full manuscript please, as an example lines 45, 49, 51, 52, 54, 94, 114 etc.
- please use most recent references: for example Atlas of MS 2013 instead of WHO 2008, EBV and MS Fernández-Menéndez 2016 instead of Lucas & Taylor 2012?
- line 123: “ Among these studies, 4 were conducted in Iran, 4 in America, 3 in British, 3 in Norwegian, 2 in Canada, 3 in Sweden, 1 in Brazil, 1 in Greek, 2 in Australia, 1 in Netherland, 1 in Kuwait, 3 in Italy, 1 in Serbia”. Please use names of the country and not language or people. So it is conducted in Norway in stead of Norwegian, United kingdom or England, Greece, The Netherlands.
- line 139-142. Study Jafari et al is excluded, but it provides information about smoking and MS within genders (table 2 of their paper) and surprisingly here the females have a higher risk. Can the authors explain why this study has been excluded?
- line 143-144: smoking in men is more dangerous than women. This is a firm conclusion, although the heterogeneity between the groups is great (p=0,001). could it be explained by age difference or other characteristic differences between the male and female studies? Please make a more considered conclusion.
- line 187: “ This finding indicates that the risk of MS has been currently rising in a much larger population.” The authors can not conclude this by the fact that between 2013 and 2015 were more studies/ publications on this subject. Last few years, there is more interest and attention for environmental factors and so more and larger studies. This does not mean that the risk is rising.
- line 190 ”MS is more common with the increasing latitude..” and not increases
- line 202: it is now more than 100 other genes next to HLA (authors mention more than 12). please use recent reference such as nice review of Sawcer in The Lancet Neurology july 2014
- Figure 1
# “ articles contained the same population WERE excluded” and not was
# the 3 excluded boxes: make the way of writing more uniform
- Table 1 check please the layout (cases, lower and upper column, make the word fit in one line)

---

## Round 0.3 · accepted · Accept

Dear authors,

I have analyzed your answers for the indicated comments and they are correct. In other words, your paper has high standards to be published in PeerJ.

Congratulations!

With respect and warm regards,
Dr Palazón-Bru